# *Trichoderma*: Harzianum Clade in Soils from Central and South America

**DOI:** 10.3390/jof10120813

**Published:** 2024-11-23

**Authors:** Adnan Ismaiel, Prashant P. Jambhulkar, Parimal Sinha, Dilip K. Lakshman

**Affiliations:** 1Environmental Microbial & Food Safety Laboratory, Agricultural Research Service, United States Department of Agriculture, Beltsville, MD 20705, USA; ed.ismaiel@usda.gov; 2Department of Plant Pathology, Rani Lakshmi Bai Central Agricultural University, Jhansi 284003, India; ppjambhulkar@gmail.com; 3Division of Plant Pathology, ICAR-Indian Agricultural Research Institute, New Delhi 110012, India; sinhapath@gmail.com; 4Molecular Plant Pathology Laboratory, Agricultural Research Service, United States Department of Agriculture, Beltsville, MD 20705, USA

**Keywords:** biocontrol agent, biofertilizer, sustainable agriculture, *Trichoderma* species, Harzianum clade

## Abstract

As environmental and health concerns increase, the trend toward sustainable agriculture is moving toward using biological agents. About 60% of all biological fungicides have *Trichoderma* species as the active ingredient, with *T. harzianum* as the most common species in these products. However, the name *T. harzianum* has often been used incorrectly in culture collections, databases, and scientific literature due to the division of the Harzianum clade (HC) into more than 95 cryptic species, with only one being named *T. harzianum*. In this study, 49 strains previously identified as *T. harzianum* in three surveys of *Trichoderma* species from soils in South and Central America were re-identified using phylogenetic analyses based on *tef1α*, *rpb2*, and ITS sequences obtained from GenBank. These were combined with the HC species from two other studies, which were identified based on the current taxonomy. Based on the results of the five surveys of the total 148 strains in HC, 11 species were identified. *T. afroharzianum, T. lentiforme*, and *T. endophyticum*, followed by *T. azevedoi* and *T. harzianum*, were the dominant species of the HC in South and Central America. This is the first report to identify dominant *Trichoderma* species within the HC in South and Central American soil based on multiple studies. These results will be useful in selecting strains within the clade for the formulation of biocontrol and biofertilizer products on the continent.

## 1. Introduction

A significant number of investigations have documented the use of beneficial microbes for disease suppression and plant growth enhancement. Species in the genus *Trichoderma* stand out for these plant-beneficial activities. *Trichoderma* as a genus was introduced in 1794 by Persoon [1]. The importance of *Trichoderma* in agriculture, specifically as a biocontrol agent (BCA) against fungal plant diseases, has been known since the 1930s [2]. Then, in the 1980s, studies showed growth promotion of various crops by application of *Trichoderma* species [3,4]. More recently, *Trichoderma* is being employed also in environmental remediation processes [5,6].

However, only in the 1990s did commercial products with *Trichoderma* as an active ingredient become commercially available with reasonable success [7]. One of the most common species in those products is *Trichoderma harzianum*. Samuels and Hebbar [1] assembled a list of commercial *Trichoderma* biocontrol products that have *T. harzianum* as the active ingredient in 21 out of 55 products, which was higher than any other *Trichoderma* species, reflecting the importance of the species for biocontrol. Also, in a compiled list of publications, Zin and Badaluddin [8] showed investigations involving the effectiveness of *Trichoderma* species against fungal crop pathogens. Within the list, *T. harzianum* was the most studied species (11 out of 18) and showed high effectiveness against various crop diseases. Taxonomically, *T. harzianum* was only one of the nine aggregate species described by Rifai [9]. Aggregate species, per Rifai, means a group of more than one species that are morphologically identical but biologically different. Taxonomy based on DNA sequencing of specific markers started in the late 1990s, which resulted in an exponential expansion of the number of species in the genus *Trichoderma*. The species, morphologically identified as *T. harzianum*, appeared to split into different clades. Those clades, in some cases, were marked by Roman numerals or Arabic numbers [10,11], without any coordination in numbering. These studies clearly showed that *T. harzianum* could represent several species that are morphologically indistinguishable. Therefore, the concept Harzianum clade (HC) started to replace *T. harzianum*. In 2015, Chaverri et al. [12] accepted 14 species within the HC, including a few that were already described. The number of species in the HC continued to expand [13,14,15,16,17,18]. Unfortunately, the split of HC did not resolve the confusion about the name completely. There are many sequences for strains deposited in databases including, GenBank as *T. harzianum*, even though *T. harzianum* is only one of the uncommon species among more than 95 described species within the clade, limiting the full value of the databases. There is another problem with the HC species. HC boundaries are not clearly identified, and mistakes happen when other species are included in the HC even though they are phylogenetically positioned outside the clade. For example, Chaverri et al. [12] did not include *T. tawa*, *T. tomentosum*, and *T. velutinum* within the HC. However, Zheng et al. [15] included all three species within the clade.

There are several surveys exploring *Trichoderma* in the soil in different parts of the world that have reported species in HC as *T. harzianum.* In this study, strains identified as *T. harzianum* in three survey studies for the isolation of *Trichoderma* strains from the soil of South and Central America were re-analyzed phylogenetically based on the available sequencing data of three loci: translation elongation factor 1α (*tef1α*), RNA polymerase subunit II (*rpb2*), and the internal transcribed spacers (ITS) obtained from GenBank. These three loci have been recommended for the identification of *Trichoderma* species [19]. After re-identification, the number of strains for each species was added to the numbers of respective species in another two studies where species in the HC were identified according to the current taxonomy. This was carried out to determine the dominant soil species of the HC on the continent. Knowing the exact dominant *Trichoderma* species in the soil from a given geographical region could help biocontrol investigations. It would also facilitate the identification of indigenous species that compete well in the soil and may have the ability to establish endophytic relationships with plants, resulting in better exploitation of plant beneficial activities by the *Trichoderma* species.

## 2. Materials and Methods

### 2.1. Evaluating the Accuracy of Trichoderma harzianum Strains Deposited in GenBank

To evaluate the accuracy of the identification of *T. harzianum* strains deposited at the National Center for Biotechnology Information (NCBI) GenBank (https://www.ncbi.nlm.nih.gov/genbank/), a search was carried out on 10 June 2024 for “*Trichoderma harzianum tef1α*” in the GenBank. The initial 100 sequence hits were downloaded in the FASTA file format. Sequences of sixteen ex-type strains in HC retrieved from GenBank were then added to this file as references. The FASTA file was aligned using Clustal Omega, version 1.2.4 (https://www.ebi.ac.uk/jdispatcher/msa/clustalo, accessed on 10 June 2024) and adjusted manually using the software Mesquite version 3.81 [20]. Then after, the file was used to construct phylogenetic trees using two methods: (1) A parsimony tree was obtained using PAUP version 4.0a (https://phylosolutions.com/paup-test/, accessed on 10 June 2024). The tree was produced using a heuristic search with a starting tree obtained by 1000 random stepwise additions of sequences, tree-bisection-reconnection (TBR) as the branch-swapping algorithm with MULTREES in effect. Gaps were treated as missing characters. Supports for branches were assessed with 1000 replicates of bootstrap. (2) A maximum likelihood tree was obtained using MEGA X, version 11.0.10 with the substitution model predetermined using MEGA X [21]. Support for the clades was assessed with 1000 bootstrap replicates.

### 2.2. Evaluation of the Dominant Species of the Harzianum Clade from South and Central America

To determine the dominant species of the HC in soils in South and Central America, strains identified based on *tef1α* in the studies of Hoyos-Carvajal et al., Smith et al., and Druzhinina et al. [22,23,24] as *T. harzianum* (49 strains) were re-analyzed phylogenetically based on the DNA sequencing data of three loci, *tef1α*, *rpb2*, and ITS. The sequences for each locus were downloaded from the GenBank and aligned with reference sequences, particularly for the ex-type specimens of known species in the clade described in Chaverri et al. [12] and del Carmen et al. [14] using the Clustal Omega, version 1.2.4 (https://www.ebi.ac.uk/jdispatcher/msa/clustalo, accessed on 10 June 2024). All the strains used in the phylogenetic analysis are listed in Table 1. The alignment files for the three genes were concatenated and adjusted visually using the software Mesquite version 3.81 [20]. The alignment file was used to construct phylogenetic trees as described above. The trees obtained by both methods were essentially identical in topology, and thus only the parsimony tree constructed by PAUP is presented.

### 2.3. Tabulation of HC Species from Soil in South and Central America

After re-identification of the strains of HC from the three studies, as detailed in Section 2.2, the number of strains of each species was added to the number of corresponding species from two other studies by Inglis et al. and Barrera et al. [25,26], which had identified nearly 100 isolates in the HC according to the correct taxonomy. The results were tabulated to determine the most prevalent species within the HC in soils of South and Central America, based on the total 148 strains of HC.

## 3. Results

### 3.1. Re-Identification of the Trichoderma harzianum Strains Deposited in GenBank

Initially, we searched GenBank for “*Trichoderma harzianum tef1α*” and phylogenetically analyzed the first 100 sequences. Based on the phylogenetic tree (Figure 1), only 22 out of 100 sequences clustered in fully supported clade (C1) with the ex-type strain of *T. harzianum* (CBS 226.95, AF348101); therefore, these strains are being identified as *T. harzianum.* Among the remaining 78 strains, many of them were recognized as other species in the HC as they fit into clades C2–C7 which include the type strains of the following species: *T. rifaii*, *T. xixiacum*, *T. guizhouense*, *T. afroharzianum*, *T. atrobrunneum*, and *T. rugulosum*, respectively. There were other strains that match into the HC but could not be identified as any known *Trichoderma* species of that clade. Two strains with accession numbers OQ200374 and OQ200375 did not belong to the HC and were identified as *T. virens*.

**Figure 1 jof-10-00813-f001:**
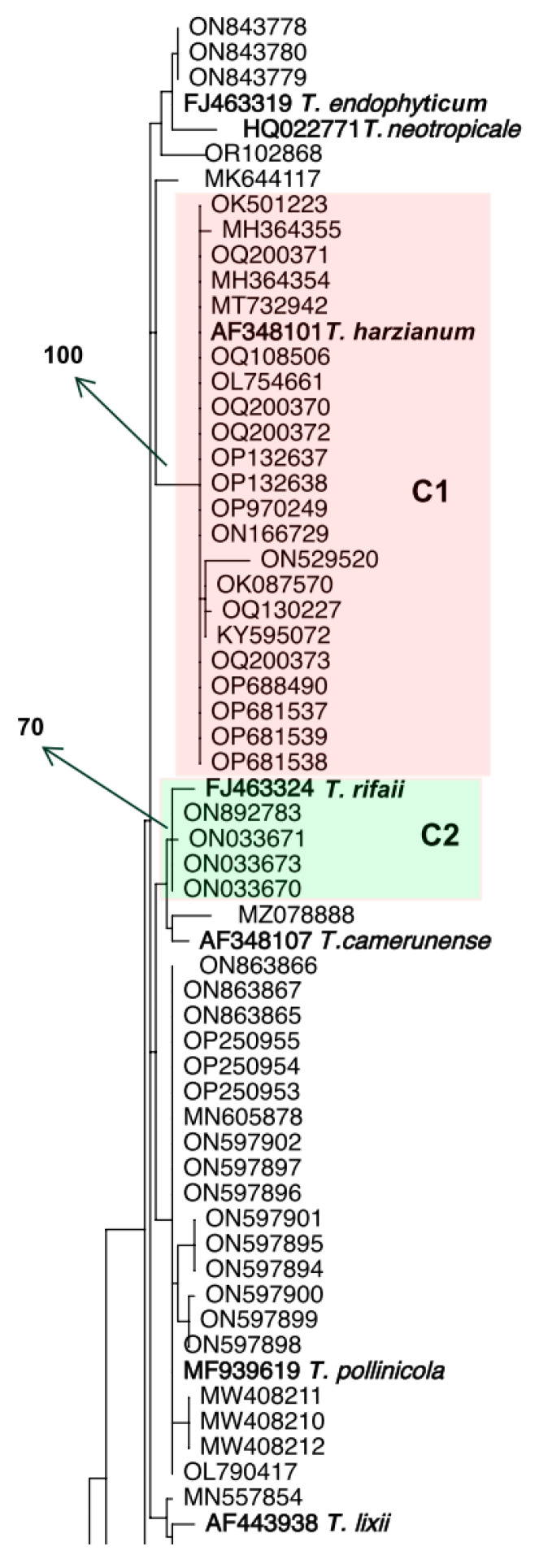
Phylogenetic tree based on *tef1α* sequence data for 100 strains retrieved from GenBank deposited as *Trichoderma harzianum*. The leaves are identified by GenBank accession numbers. Numbers given above the branches indicate bootstrap values of ≥70% obtained via 1000 replications. The boldface indicates type strains included to identify the clades. Colored clades C1–C7 indicate strains clustered with a type strain of an HC species with a bootstrap value of ≥70%. The tree was rooted to an ex-type specimen of *T. virens*.

### 3.2. Evaluation of the Dominant Trichoderma Species of the Harzianum Clade from South and Central America

In the three survey studies by Hoyos-Carvajal et al., Smith et al., and Druzhinina et al. [22,23,24] for isolation of *Trichoderma* species in the soil of Central and South America, all the strains in the HC were reported as *T. harzianum*, which does not reflect the split of the clade into more than 95 species. Phylogenetic re-analyses of sequencing data (*tef1α*, *rpb2*, and ITS) of the 49 strains (Figure 2) showed that 25 strains clustered with the type strain of *T. lentiforme* in clade C1 with a high bootstrap value of 79 and are recognized here as belonging to that species.

In clade C2, 10 strains from the three studies formed a highly supported clade with the ex-type strain of *T. afroharzianum* and two other reference sequences of strains previously identified as *T. afroharzianum* supporting the identification of these strains as *T. afroharzianum*.

In clade C3, one isolate from Peru, CIB T52 clustered with three reference strains identified as *T. pyramidale* and *T. pseudopyramidale.* However, the CIB T52 is closer to *T. pseudopyramidale* than to *T. pyramidale.* These two species are closely related, but *T. pseudopyramidale* has been found in Africa and the other in South America [12,14].

The clade C4 included six strains together with the type strains of *T. endophyticum*, *T. neotropicale*, and *T. afarasin*. The type strain of *T. neotropicle* seems to be distantly related to the clade through a long branch, and *T. afarasin* is known to be an African species not found outside that continent. Therefore, it is most likely that the six isolates in the clade were *T. endophytcum*.

In clade C5, two strains CIB T100 and TUB-F1078 from two different studies clustered with the type strain of *T. harzianum* (CBS 226.95) with a high bootstrap value (BS = 99) and were recognized as *T. harzianum*.

In clade C6, seven strains nested with the ex-type strain of *T. azevedoi* (CEN1422); in most cases the sequences were identical. In the study of Barrera et al. [26], five strains from Argentina were identified as *T. austroindianum*. Four of the five strains have identical *tef1α* sequences. The other strain has 1 gap difference with the other four isolates. The type strain of *T. austroindianum* had sequences highly homologous to those of *T. azevedoi* strains and fell in clade 6 with *T. azevedoi*, and thus, we believe that the two species represent only one species. Accordingly, the strains of *T. austroindianum* were tabulated with the *T. azevedoi* as, on a priority basis, the latter species was described before *T. austroindianum* [27].

In clade C7, one isolate from Colombia, CIB T99, formed a clade with the type strain of *T. rifaii* and another reference strain of *T. rifaii* with a BS value of 71, suggesting the correct identification of the strain as *T. rifaii*.

The strain DAOM 234005 did not fit into any clade representatives of the known species of the Harzianum clade. Moreover, based on BLAST search, the *tef1α* sequence of this strain had no close homologous sequence to it in the GenBank; the nearest sequence to it was accession number KU238051 with a genetic distance of 0.0578 as determined by PAUP. This accession number belongs to a strain (TS187) from Malaysia deposited in GenBank as *Trichoderma* sp. [28]. Thus, we consider this strain as a possible new species in the HC.

A previous study [25] described 12 isolates from Brazil as *T*. *peberdyi* belonging to HC. However, the phylogenetic tree (Figure 2) showed that *T. peberdyi* is positioned between three species: *T*. *tomentosum* (outgroup), *T. pleuroti*, and *T. pleuroticola*, all of which were considered outside of HC [12].

The results of our re-identification of species of HC combined with the previously identified species in two other studies [25,26] are presented in Table 2 and summarized in Figure 3. Overall, strains of 11 species of HC were isolated in soils of South and Central America. *T. afroharzianum* was the most common species in the region, with 44 strains found in four out of five studies. This is a well-known biocontrol species, and the strain T22 is an example [7]. *T. lentiforme* was the second most common species, with 39 strains from four out of five studies. The strains of both species were distributed in all the regions from north to the south of the continent.

## 4. Discussion

Biocontrol programs are in place in South and Central America, notably in nations like Brazil, Argentina, Colombia, and Mexico, to manage plant diseases and encourage crop development with biological agents [29]. One of the most important fungal names in the field of biocontrol products for plant diseases and/or plant health promotion are species in the genus *Trichoderma* and, in particular, *T. harzianum* [1,12,30]. However, confusion exists about the name *T. harzianum* being used for all the species within the Harzianum clade, despite the revision of the phylogeny of the clade and the description of more than 95 species that started in 2015 [12]. In fact, the problem is widespread and persistent in scientific publications, databases, and commercial products. As an example, after searching the GenBank for “*Trichoderma harzianum tef1α*” and identifying the first 100 sequences, this study revealed that only 22 clustered with the ex-type of the *T. harzianum* strain (GenBank accession number AF348101) and could be identified as *T. harzianum*, indicating 78% misidentification in the name of strains deposited as *T. harzianum*. What is disturbing about this is the fact that 100% of the sequences used in Figure 1 were deposited between 2018 and 2023, at least three years after the major revision of the HC [10]. Precise naming of species of HC is critical as beneficial properties of biocontrol and plant growth promotion are species-specific or even strain-specific. Due to improper usage of the name, there is a lack of studies exploring which species in the clade are dominant species in soil, despite the HC species being commonly isolated in many survey investigations. Dominance in the soil is an important criterion for selecting any biocontrol strain, as it reflects the fungus’s high ability to compete, outgrow, and suppress other species, leading to a higher potential for the fungus to establish endophytic relationships with plants and induce systemic resistance to pathogens. Through inducing systemic resistance, the *Trichoderma* species provides benefits other than disease resistance, such as growth promotion, resistance to abiotic stress, and high efficiency in using nitrogen [7].

In this study, we re-analyzed the data from three survey studies published in 2005–2013 using multi-locus phylogeny. The number of re-identified species (49) was added to their respective species from the two other studies where HC species were identified, taking into account the split of the clade [22,23,24,25,26]. *T. afroharzianum* and *T. lentiforme*, respectively, were the top two dominant species on the continent and were obtained from north to south. *T. afroharzianum* is cosmopolitan and a well-known species of biocontrol agents, and the *T. afroharzianum* strain T22 is well known as a biocontrol and biofertilizer strain [31]. *T. lentiforme* was reported to be mainly an endophytic fungus [12]. The dominance of *T. lentiforme* in soil is a new report of our findings. Having this ability in soil and the ability to establish endophytic relations with plants are top criteria for selection with biocontrol and biofertilizer properties. Recently, *T. lentiforme* was found to have biocontrol activity against *Sclerotinia sclerotiorum* and had growth stimulant properties in cotton [32].

*T. endophyticum* and *T. azevedoi* are the third and fourth most common species, respectively, on the continent. *T. endophyticum* was shown to be exclusively endophytic based on the strains available in a previous study [12]. However, this study showed that this species is a soil fungus as well (Table 2). *T. azevedoi* was described in the study of Inglis et al. [25]. However, the strains are not geographically restricted to the continent of South America, as strains of this species have also been found in Australia, e.g., strain number BRIP 74284 with *tef1α* GenBank accession number of OR802290. In this context, we corrected the name of a species described by Barrera et al. [26] as *T. austroindianum.* Five strains of *T. austroindianum* appear to have identical or highly homologous *tef1α* sequences to those of *T. azevedoi*, and both clustered in one highly supported clade. Thus, we renamed all five strains of *T. austroindianum* as *T. azevedoi* in Table 1. Here we stress the importance of (1) BLAST search of the sequences of *tef1α* or *rpb2* loci of unknown *Trichoderma* strains before describing them as new species to avoid duplication of species naming and (2) to include the most homologous species to it in the phylogenic analyses. The phylogenetic trees in Barrera et al. [26] did not include *T. azevedoi* in the analyses, which is probably the reason that the authors overlooked this error.

The boundaries of the species within the HC are equally unclear. For example, Chaverri et al. [12] did not include *T. tawa*, *T. tomentosum*, and *T. velutinum* within the clade of Harzianum. However, Zheng et al. [15] included all the above three species within the clade. In another example, Chaverri et al. [12] excluded *T. amazonicum* and *T. pleuroticola* from HC. Yet Chen and Zhuang [33] placed both species inside the HC. In this regard, Inglis et al. [25] described a new species named *T. peberdyi* as part of the HC. Based on BLAST search at NCBI GenBank and our phylogenetic tree (Figure 2), this species is closely related to *T. tomentosum.* Based on Chaverri et al. [12], *T. tomentosum* is not part of the clade. Therefore, *T. perbedyi* was not considered part of the HC clade and was excluded from our list of HC species (Table 2, Figure 3). Setting a genetic distance limit within HC species could be a way to solve the boundary issue.

We also attempted to compare the dominant species in Central and South America with those of other continents. However, there is currently insufficient data that quantitatively show the number of HC strains within the total number of isolated *Trichoderma* strains. Thus, comparing South and Central America with other continents was not feasible. Nevertheless, we found reports that allowed us to make a comparison with our results. For example, a survey of *Trichoderma* isolates from the soil of India [34] showed that among 15 strains in HC identified, 11 were *T*. *afroharzianum*. These data corroborate the high prevalence of *T. afroharzianum* in the soil in South and Central America. On the other hand, a survey study from Iran showed that *T. afroharzianum* was not a dominant HC species from soil in the western region of the country [35]. In fact, *T. harzianum* was the most frequently isolated species. In another study, Tang et al. [36] showed that species of HC were the most prevalent *Trichoderma* species from soil in the region of Zoige Alpine of China. Among species of the HC, *T. harzianum* was the most prevalent, representing about 72% of the species in the HC (37/51). This finding differs from what we obtained in South and Central America. *T. harzianum* is known to be a species of cold temperature region, and the temperature range of the soil in the region where samples were obtained ranges from −10 °C in winter to 15 °C in summer [37]. This may have caused a bias in species dominance. In South Africa, du Plessis et al. [38] showed that species of *T. afroharzianum*, *T. atrobrunneum*, and *T. camerunense* were the only species within HC obtained from the soil in South Africa; however, the authors did not report the frequency of these of each species to know the most prevalent species in that region.

## 5. Conclusions

Within the genus *Trichoderma*, *T. harzianum* is one of the most well-known species for biocontrol and plant growth promotion. The Harzianum clade’s (HC) split has led to the improper usage of this species’ name in scientific publications, databases, and commercial products. This study used data from five investigations to identify *Trichoderma* species within the HC in the soil of South and Central America. The species *T. afroharzianum*, *T. lentiforme*, *T. endophyticum*, *T. azevedoi*, and *T. harzianum* were found to be the dominant species of the HC in South and Central America. However, the soil sampling in those five studies was not obtained in a statistical manner and order. Thus, we consider the findings from this study to provide a rough estimate of the dominant species of HC on the continent. Selecting strains or species for biocontrol and growth promotion from them could be an expensive and arduous undertaking. Identifying the dominant species within HC and selecting strains from them can expedite the commercialization process by reducing the time and expense associated with strain selection. Moreover, these strains may have better fitness traits than the rest of the species in HC. We further emphasize the need for additional research to be carried out in the future from different continents, as we were unable to locate any data with which to compare our findings. Moreover, we emphasize the significance of accurately identifying *Trichoderma* species prior to depositing their sequences into databases such as GenBank or culture collection institutes such as the American Type Culture Collection (ATCC).

## Figures and Tables

**Figure 2 jof-10-00813-f002:**
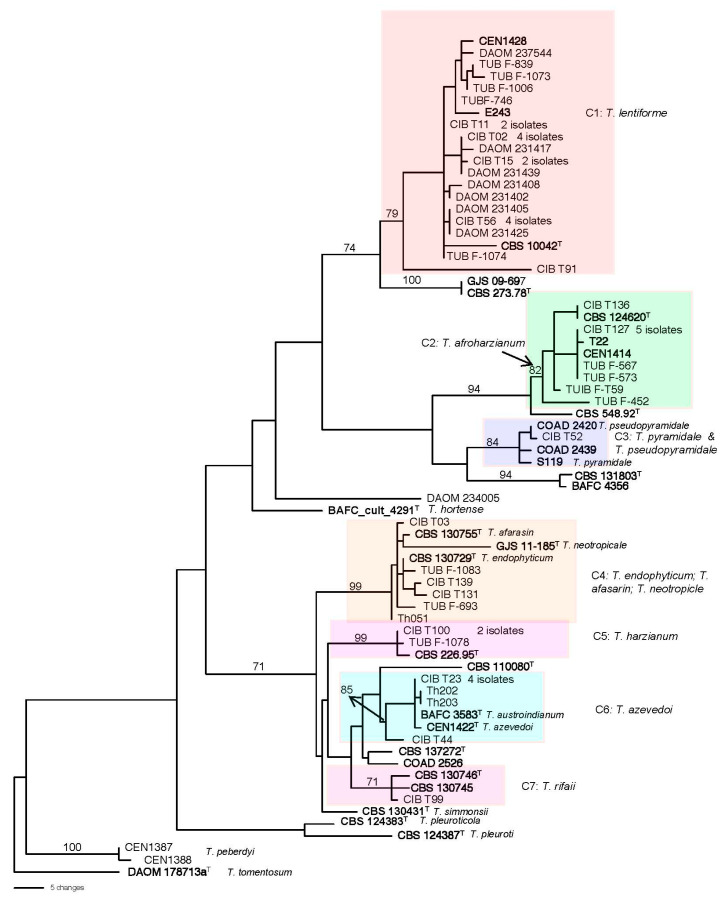
One of the most parsimonious trees generated by phylogenetic analysis of combined DNA sequences of *tef1α, rpb2*, and ITS. Numbers given above the branches indicate bootstrap values of ≥70% obtained via 1000 replications. Leaves are identified by strain numbers. Boldface indicates reference strains included to identify the clades. ^T^ After strain numbers indicate the ex-type strain. Clades (C1–C7) refer to lineages that have strains identified in this study. Tree was rooted to the type strain of *T. tomentosum*.

**Figure 3 jof-10-00813-f003:**
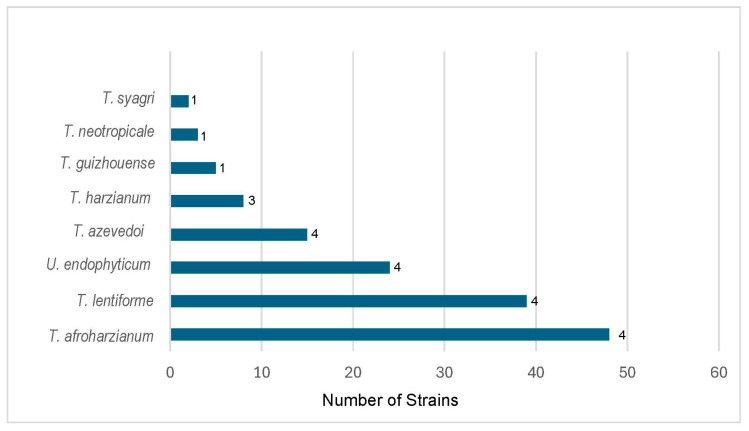
Dominant species of *Trichoderma* in the Harazianum clade in the soil in Central and South America. The numbers at the end of the bars represent the frequency of presence of a species in five studies.

**Table 1 jof-10-00813-t001:** *Trichoderma* species with their origin, strain number, GenBank accession number, and the number of strains with identical *tef1α* GenBank accession numbers included in Figure 1.

*Trichoderma* Species	Origin	Strain #	*Tef1α*	ITS	*rpb2*	No. of Strains ^b^
***T. afarasin* ^a^**	**Cameroon**	**CBS 130755 ^T^**	**AF348093**	**AY027784**	**-**	
*T. afroharzianum*	Colombia	CIB T136	EU279981	EU280078	-	
	Colombia	CIB T127	EU279980	EU280078	-	5 ^b^
	**Colombia**	**T22**	**AF469194**	**AF469188**	**-**	
	**Brazil**	**CEN1414**	**MK696652**	**MK714894**	**MK696813**	
	**Peru**	**CBS 124620 ^T^**	**FJ463301**	**FJ442265**	**FJ442691**	
	Colombia	CIB T59	EU279986	EU280078		
	Peru	TUB F-567	AY857267	AY857208	-	
	Peru	TUB F-452	AY857266	AY857206	-	
	Peru	TUB F-573	AY857268	AY857209	-	
** *T. atrobrunneum* **	**France**	**CBS 548.92 ^T^**	**AF443942**	**AF443924**	**-**	
** *T. austroindianum* **	**Argentina**	**BAFC 3583**	**MH352421**	**-**	**-**	
*T. azevedoi*	Colombia	CIB T23	EU279989	EU280077	-	
	Colombia	Th202	AB558911	-	AB558921	
	Colombia	Th203	AB558912	-	AB558922	
	Colombia	CIB T44	EU279983	EU280077	-	
	**Brazil**	**CEN1422**	**MK696660**	**MK714901**	**MK696821**	
** *T. botryosum* **	**Ethiopia**	**COAD 2526**	**MK044147**	**-**	**MK044240**	
** *T. camerunense* **	**Cameroon**	**CBS 137272 ^T^**	**AF348107**	**AY027780**	**-**	
*T. endophyticum*	Colombia	CIB T03	EU279977	EU280079	-	
	**Ecuador**	**CBS 130729 ^T^**	**FJ463319**	**FJ442243**	**-**	
	Guatemala	TUB F-693	AY857271	AY857211	-	
	Mexico	TUB F-1083	AY857300	AY857253	-	
	Colombia	CIB T139	EU279991	EU280075	-	
	Colombia	Th051	AB568382	-	AB568476	
	Colombia	CIB T131	EU279988	EU280075	-	
** *T. guizhouense* **	**China**	**CBS 131803 ^T^**	**JN215484**	**JN191311**	**JQ901400**	
	**Argentina**	**BAFC 4356**	**MG797485**	**-**	**-**	
*T. harzianum*	Colombia	CIB T100	EU279978	EU280079	-	
	**U.K.**	**CBS 226.95 ^T^**	**AF348101**	**AJ222720**	**AF545549**	
	Mexico	TUB F-1078	AY857298	AY857250	-	
** *T. hortense* **	**Argentina**	**BAFC_cult_4291 ^T^**	**MH253895**	**-**	**-**	
** *T. inhamatum* **	**Peru**	**G.J.S. 09-697**	**KP115272**	**-**	**-**	
	**Colombia**	**CBS 273.78 ^T^**	**AF348099**	**FJ442680**	**FJ442725**	
*T. lentiforme*	Brazil	TUB F-746	AY857257	AY857216	-	
	Colombia	CIB T11	EU279979	EU280079	-	2
	Colombia	CIB T02	EU279976	EU280079	-	4
	Mexico	DAOM 231417	AY605771	AY605728	-	
	Colombia	CIB T15	EU279982	EU280079	-	2
	Mexico	DAOM 231408	AY605773	AY605730	-	
	Mexico	DAOM 231402	AY605775	AY605732	-	
	Mexico	DAOM 231439	EU279994	AY605728	-	
	Colombia	CIB T91	EU279987	EU280079	-	
	Mexico	DAOM 231405	AY605774	AY605731	-	
	Colombia	CIB T56	EU279985	EU280079	-	4
	Peru	DAOM 237544	EU279993	EU280133	-	
	Mexico	DAOM 231425	AY605768	AY605725	-	
	**Cameroon**	**E243**	**MK044089**	**-**	**MK044182**	
	Mexico	TUB F-839	AY857283	AY857231	-	
	Brazil	TUB F-1073	AY857295	AY857247	-	
	Brazil	TUB F-1006	AY857286	AY857235	-	
	**French Guiana**	**CBS 100542 ^T^**	**AF469195**	**AF469189**	**-**	
	Argentina	TUB F-1074	AY857296	AY857248		
	**Brazil**	**CEN1428**	**MK696667**	**MK714909**	**MK696827**	
** *T. lixii* **	**Thailand**	**CBS 110080 ^T^**	**AF443938**	**AF443920**	**-**	
** *T. neotropicale* **	**Peru**	**G.J.S. 11-185 ^T^**	**HQ022771**	**HQ022407**	**-**	
** *T. peberdyi* **	**Brazil**	**CEN1387**	**MK696619**	**MK714861**	**MK696781**	
	**Brazil**	**CEN1388**	**MK696620**	**MK714862**	**MK696782**	
** *T. pleuroti* **	**South Korea**	**CBS 124387 ^T^**	**HM142382**	**HM142363**	**HM142372**	
** *T. pleuroticola* **	**South Korea**	**CBS 124383 ^T^**	**HM142381**	**HM142362**	**HM142371**	
** *T. pseudopyramidale* **	** *Ethiopia* **	**COAD 2420 ^T^**	**MK044115**	**-**	**MK044208**	
	** *Ethiopia* **	**COAD 2439**	**MK044171**	**-**	**MK044264**	
	Peru	CIB T52	EU279984	EU280077		
** *T. pyramidale* **	** *Italy* **	**S119**	**KJ665696**	**-**	**-**	
** *T. rifaii* **	**Ecuador**	**CBS 130746 ^T^**	**FJ463324**	**FJ442663**	**-**	
	**Panama**	**CBS 130745**	**FJ463321**	**FJ442621**	**FJ442720**	
	Colombia	CIB T99	EU279990	EU280103	-	
** *T. simmonsii* **	**USA, MD**	**CBS 130431 ^T^**	**AF443935**	**AF443917**	**FJ442757**	
*Trichoderma* sp.	Peru	DAOM 234005	EU279992	EU280091	-	
** *T. tomentusom* **	**Canada**	**DAOM 178713a ^T^**	**AY750882**	**EU330958**	**AF545557**	

^a^ boldface strains are reference sequences; ^b^ numbers in this column indicate more than one strain had the same *tef1α* GenBank accession number; ^T^ ex-type specimen.

**Table 2 jof-10-00813-t002:** *Trichoderma* species in the Harzianum clade in soils of South and Central America based on data of five survey studies.

Species ^a^	Strain Number	Number of Isolates	*tef1α* Accession Number	Country	Ref.
*T. afroharzianum*	CIB T136	1	EU279981	Colombia	[22]
	CIB T07 CIB T63 CIB T61 CIB T53 CIB T127	5	EU279980 *	Colombia	[22]
	CIB 59	1	EU279986	Colombia	[22]
	CEN1410 CEN1414 CEN1417	3	MK696648 *	Brazil	[25]
	TUB F-567 TUB F-573 TUB F-452	3	AY857267 AY857268 AY857266	Peru	[24]
	BAFC 4374 BAF 4392 for the rest see the reference	35	MH395411, MH395415	Argentina	[26]
Total		48			
*T. lentiforme*	DAOM 237544	1	EU279993	Peru	[22]
	CIB T02 CIB T112 CIB T35 JB M10-2	4	EU279976 *	Mexico, Colombia	[22]
	CIB T15, CIB T41	2	EU279982 *	Colombia	[22]
	DAOM 231417	1	AY605771	Mexico	[22]
	DAOM 231439	1	EU279994	Mexico	[22]
	DAOM 231408	1	AY605773	Mexico	[22]
	DAOM 231405	1	AY605774	Mexico	[22]
	DAOM 231425	1	AY605768	Mexico	[22]
	CIB T56 CIB T60 CIB T16 DAOM 229985	4	EU279985 *	Panama, Colombia	[22]
	CIB T91	1	EU279987	Colombia	[22]
	CIB T11 CIB T102	2	EU279979 *	Colombia	[22]
	DAOM 231402	1	AY605775	Mexico	[22]
	CEN1412 CEN1415 CEN1416 CEN1428 CEN1429	5	MK696650 MK696653 MK696654 MK696668 MK696667	Brazil	[25]
	TUB F-839 TUB F-1073 TUB F-1006 TUB F-746 TUB F-1074	5	AY857283 AY857295 AY857286 AY857257 AY857296	México Brazil Brazil Brazil Argentina	[24]
	BAFC 4391 BAFC 4394 for the rest see the reference	9	MH036883 MH036885	Argentina	[26]
Total		39			
*T. endophyticum*	CIB T03, CIB T131, CIB T139	3	EU279977, EU279988, EU279991	Colombia	[22]
	TUB F-1083, TUB F-693	2	AY857300, AY857271	Mexico, Guatemala	[24]
	Th051	1	AB568382	Colombia	[23]
	BAFC 4358, BAFC 4372 for the rest see the reference	18	MH371393 MH371397	Argentina	[26]
Total		24			
*T. azevedoi*	CIB T23 CIB T24 CIB T126 CIB T128	4	EU279989 *	Colombia	[22]
	CIB T44	1	EU279983	Colombia	[22]
	CEN1422 CEN1423 CEN1403	3	MK696660 MK696661 Mk696638	Brazil	[25]
	Th202 Th203	2	AB558911 AB558912	Colombia	[23]
	BAFC 3583 BAFC 3844 GJS 08-128 GJS 08-181 VAB-T051	5	MH352421 MG822709 MH352423 MH352422 MH352424	Argentina	[26]
Total		15			
*T. harzianum*	CIB T100 PER4-2	2	EU279978 *	Colombia Peru	[22]
	TUB F-1078	1	AY857298	Mexico	[24]
	GJS 08-172 GJS 08-173 VAB-T032 VAB-T052 VAB-T053	5	KT275197, KT275198, KT275199, MH364354, MH364355	Argentina	[26]
Total		8			
*T. guizhouense*	BAFC 4356 BAFC 4370 GJS 08-102 GJS 08-121 VAB-T047	5	MG797485 MG797486 MG797484 MG797483 MG797482	Argentina	[26]
*T. neotropicale*	GJS 08-182 GJS 08-183 VAB-T049	3	MG822718, MG822719, MG822720	Argentina	[26]
*T. syagri*	BAFC 4357 BAFC 4371	2	MG227714, MG227710	Argentina	[26]
*T. pseudopyramidale*	CIB T52	1	EU279984	Peru	[22]
*T. hortense*	GJS 08-116	1	MH253895	Argentina	[26]
*T. rifaii*	CIB T99	1	EU279990	Colombia	[22]
*Trichoderma* sp.	DAOM 234005	1	EU279992	Peru	[22]

^a^ species arranged from the most to the least dominant; * strains have identical *tef1α* GenBank accession numbers.

## Data Availability

The original contributions presented in the study are included in the article, further inquiries can be directed to the corresponding author.

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
