# Peer review of "Trichoderma: Harzianum Clade in Soils from Central and South America"

_jof, 2024, doi:10.3390/jof10120813_

Round 1

Reviewer 1 Report (Previous Reviewer 3)

The authors responded to my suggestions. The version provided by the authors is suitable for publication.

No comments

Author Response

Reviewer has no comments

Reviewer 2 Report (New Reviewer)

The topic of this paper is interesting and has a great importance in terms of biocontrol applications not only in Central and South America but on other continents, as well. The text is written with perfect English, it is informative and easily digestible. The methods are adequate, the results are clear and the conclusions are logical and moderate.

In my opinion it was a good idea to upload the paper to the preprints.org. Though, publicly were not presented, I guess that some helpful comments were sent to the authors that were included in the final version of the manuscript.

The rate of self-citation, including both the authors and MDPI journals, is very low: Jambhulkar (ref.33); Journal of Fungi (ref.15 and 18).

Similarity degree is very low, see attached file.

I missed a bit the list of those 100 GenBank sequences that were used for analysis. Maybe a supplementary file, containing the IDs, could be added to the document.

Please, check the sentence in line 289-290 and make a correction (typos).

Some minor corrections:

- check tef1a (sometimes is written as Tef1a; e.g. Table 1 caption, line 166)

- clade is written as "Clade" in line 172

- line 234: However, Three (change "T" to "t".

See above comments.

Author Response

Reviewer has no comments

Reviewer 3 Report (New Reviewer)

To know the dominant Trichoderma species of the Harzianum clade in South and Central America with potential use as biocontrol agents, this study is focused on the reidentification of Trichoderma harzianum strains with sequences deposited in Genbank database. Those strains were from three soil surveys conducted in the mentioned areas between 2005-2013 with results published in Druzhina et al. (2005), Hoyos-Carvajal et al. (2009) and Smith et al. (2013). Sequences of gene markers (tef1α, rpb2, and ITS) currently recommended for species identification in Trichoderma of 49 strains were retrieved from the GenBank and analyzed phylogenetically in the present study. To obtain confinable and robust results, the authors include in the analyses sequences of the type strains of the species of the Harzianum clade and reference strains of species collected in South and Central America and molecularly identified by Barrera et al. (2021) and Inglis et al. (2020).

Sequence revision and species reidentification of strains accessed in public databases should be continuous work to update strain identification based on the latest taxonomy insights for a concrete genus or group of fungi. Therefore, this study is welcome because it contributes significantly to updating the identification of a big set of T. harzianum strains through available sequences in Genbank, revealing that they belonged to 11 different species of the Harzianum clade. However, the manuscript's content is excessively confusing, repetitive, and riddled with errors that can cause further confusion to the reader. In addition, the submitted version seems to be provisional rather than a final version due to the number of words, sentences, and paragraphs found in yellow-red colours. Therefore, the authors should review the present version and modify its content taking into account the recommendations listed below if they agree.

In the Abstract and, at least, in the last paragraph of the Introduction (objectives), it must be well stated that the study is based on Genbank sequences of T. harzianum strains from the surveys mentioned above. The publications of those surveys should be explicitly cited in this latter part. In the present version, the reader knows that only when reading two paragraphs, one in results (Ln152) and the other in the discussion (Lns 273-276). Be careful, in these lines the authors introduced the references 19 and 20, but these are articles concerning basic methods for sequence analyses. Consider replacing them with 22 and 23, which articles are more related to the context of the sentence! Another mistake is the number of strains studied; while in Discussion it is 48, in other parts of the ms is 49. Please check!

Regarding some expressions in the text, it makes you think that the authors have not too much experience in taxonomy. For instance, they do not distinguish between a reference strain from an ex-type strain of species, or they do not know the meaning of "type species” because they mix all these terms. The authors must know that a "type species” exclusively refers to the species representative of a genus; the "type strain" is the specimen representative of a species (ex-type strain = a culture from the type strain), and a reference strain is a well-identified strain of a species, but different to the (ex-)type strain of the same species. All those reiterative confusions must be corrected along the text in the new version of the ms, including Figure legends.

The proposal of the authors “Harzianum Complex Clade” (Lns 57-58) can induce more confusion than clarification for the reader. In taxonomic Trichoderma studies, these terms, “clade” and “complex”, are used separately since they have different meanings. In the recent literature (Cai & Druzhinina 2021, etc), the most widely accepted term is “Clade”, which refers to a group of species that share a common ancestor and is represented by a well-supported lineage in a phylogenetic tree. Therefore, the authors should respect this concept and replace their proposal for the Harzianum Clade (HC or THC) like other Trichoderma clades representatives of phylogenetically related species. All the text must be modified accordingly, including the title (see suggestion below)

The authors based their reidentification of the strains on the three gene markers that Cai & Druzhinina (2021; https://doi.org/10.1007/S13225-020-00464-4) recommended in the widely accepted protocol proposed for an accurate identification of Trichoderma species. However, it is surprising that this publication is not among the references of the present study. It is the most recent and relevant taxonomic study on Trichoderma and the authors must consider introducing this publication in the ms. Cai & Druzhinina emphasize the use of tef1α, rpb2, and ITS sequences for identification and species delineation in the genus. A weak point in the present study is the lack of rpb2 sequences from the strains (Table 1) that are intended to be reidentified, and this significantly limits the reliability of identification. So, Be careful with the identification of T. endophyticum or T. azevedoi since rpb2 is not available for practically any American strain, including for the type strains of the respective species. The set of strains under these names could be also identified as T. afarasin or T. austroindicum. The conclusion included as explanations of these two identifications (Lns 79-82 and 192-200) are not very convincing. Pay attention with the epithet of T. azevedoi since in the ms it is several times incorrectly written as “azedevio”, ”azevvedoi”.

Regarding phylogenetic trees, it is more appropriate to use evolutionary distances rather than changes to quantitatively represent the evolutionary relationships between species. It is also relevant to include the significant values of statistical supports associated with the branches of the trees. So, authors must replace the asterisks with values obtained by the two phylogenetic inferences (parsimony/bootstrap support). Only showing those values, you can say that a clade is weak, moderate or highly supported (in Lns 172 and 293 give support values). Both trees could be improved by indicating the clades of each species with colours and displaying species names and numbers in a larger font.

Furthermore, several numbers of strains do not correspond with the same strain mentioned in Table, Figures o in the text. Also, some numbers in Table 1 are not included in the Figures, therefore their reidentification is not proved. Please check!

Fig 1. Represent the multi-locus analysis inferred with tef1α, rpb2, and ITS. The sequence for each marker has its own accession number. Therefore, there should be three accession numbers per strain in that tree. However, there is only one. So, delete the sequence number (leave only the strain number) or, otherwise, include all three.

The discussion is reiterative regarding the information in the introduction and results. It must be shortened and modified in light of this revision's comments.

Consider modifying the title “Harzianum clade species of Trichoderma in soils from Central and South America: reidentification and dominant species determination” or “Trichoderma: Harzianum clade in soils from Central and South America.

Harzianum Complex Clade” should be deleted and/or replaced with Harzianum clade or HC along the text.

The following sentence “….another two studies for which species in the HCC were correctly identified” or similar (Lns 20, 77-78, 122) is repeated along the ms (Abstract, Introduction, M&M ), however, the authors never explain the basis on what they consider that the species were correctly identified in these studies. This must be explained at the beginning of the ms.

Lns 17-23. This paragraph is confusing because it gives rise to the impression that the authors have the strains and they get the sequences. Please clarify!

Lns 72-81. The references of the three surveys and the two studies from where sequences used ain the present analyses were published must be included in this paragraph.

Ln 83. “Secondly” it means that previously in the text there is a sentence with “Firstly” and it is not so. Modify!

Ln 87. Change title accordingly.

Ln 91. Replace “Trichoderma harzianum translation elongation factor” with “sequences of Trichoderma harzianum tef1α gene”.

Ln 107. Replace “translation elongation factor” with “tef1α

Lns 109-110. Only mention the abbreviations of the respective genes.

Ln 113. The reference to “del Carmen et al” is associated with the number 14, not 12.

Ln 152, the references [19, 20, 21] do not correspond to the citations in the preceding text. The correct references should be [21, 22, 23].

Ln 153. Delete “(49 strains)” it is not necessary.

Ln 156. “type strain” not “type species”

Ln 157. “with high bootstrap values” This sentence is incorrect since it is impossible to know since bootstrap values are not shown in any trees. Delete this information or include values in the tree.

Ln 158. “Two other reference sequences in Clade 1 were also used.” This sentence is not necessary.

Ln 158-162. Delete this information since it was explained in general before.

Lns 172-203. Delete “Figure 2” in all the paragraphs. All this part should be simplified and several comments included should be transferred to the discussion.

Ln 201, the strain CIB T52 is incorrectly mentioned as being part of clade 7, whereas it is present in clade 3. The correct strain should be CIB T99, as shown in Figure 2.

Lns 204-211. This paragraph should be better expressed. In addition, if Blast search is done, the percentage of similarity should always be given here and anywhere!!

Lns 225-245. It is Discussion. Please transfer there most of the text.

Ln 257. Replace “translation elongation factor” with “tef1α

Ln 274. Correct the misprints “strains” and “geographically”.

Lns 334-343 “…more than 95 species…” repeated here and in other parts of the text. Avoid repetitions.

Table 1.

-        The species should be listed in alphabetical order, and it is not necessary to repeat the name. For example, T. afroharzianum is repeated up to three times. Type strains should not be marked with an asterisk next to the species name; they should be shown with a T next to the strain accession number. For example, T. lentiforme "CBS 100542T". What do the two asterisks mean in T. lixii?

-        The GenBank accession number MK666796 for tef1α in T. lentiforme is incorrect. This accession belongs to a chloroplast of Pelargonium viscosissimum.

-        Review all the Table, including the accession numbers.

Table 2. Listing also the species in alphabetical order.

-        JBPER6-2 This is not in Fig 2 nor the text. Check!!

Author Response

I am attaching a file to respond

Round 2

Reviewer 3 Report (New Reviewer)

The text has been improved considerably. However, the writing quality of this new version remains inadequate, particularly in the results section, due to the frequent use of identical sentences and some confusing sentences.

I have included corrections and suggestions for the text in the attached PDF file.

Regarding Figures 1 and 2. The phylogenetic trees still need improvement. The font size of the species names in Fig. 1 is too small. There are a lot of type strains (T) without the species name in Fig. 2. Since this is very confusing for the reader, please modify accordingly.

Space in the text should be optimized.

Check citations in the text since there are mistakes. Check also the format of the list of references.

Author Response

Response to Reviewer #3 comments:

We profusely thank the reviewer for the time spent and the valuable comments. We edited the manuscript point by point as suggested by the reviewer.  Below please find specific comments of the reviewer with our response.

Comments:

The text has been improved considerably. However, the writing quality of this new version remains inadequate, particularly in the results section, due to the frequent use of identical sentences and some confusing sentences.

I have included corrections and suggestions for the text in the attached manuscript (word file) file.

Responses:

Again, we thank the Reviewer for the corrections and suggestions provided to improve the manuscript, particularly, in the results section. The manuscript has been edited and all the reviewer’s comments have been incorporated including editing, rewriting, deleting, introducing a tabulation section in the methods, and mistakes in reference number citation, etc. The attached manuscript has all those edits incorporated.

As to the question of the word “leaves”, we say that leaves mean the tips of the tree where we put labels such as strain number or accession number.

Comments:

Regarding Figures 1 and 2. The phylogenetic trees still need improvement. The font size of the species names in Fig. 1 is too small. There are a lot of type strains (T) without the species name in Fig. 2. Since this is very confusing for the reader, please modify accordingly.

Responses:

Figure 1- Unfortunately, the font in Figure 1 is already the largest possible for the tree. Any enlargement would not leave space for all the accession numbers. We would also say that these days, people commonly read a paper on the screen of a computer and the font can be increased and the tree will be visible clearly to all readers.

Figure 2. The “T” in this figure has been modified. When there is only one strain number with “T” in clades C1-C7, we did not add name by the “T” since the name by the clade number indicates the name for all the strains in the clade including that with “T” e.g. C1 and C2.  However, when there were multiple “T” in one clade such as clade C3 and C4, we added names for all the strain numbers with “T”. We also added names for some “Ts” outside the clades C1-C7 since they have been mentioned in the text and they are important to be identified.

Comments:

Space in the text should be optimized.

Check citations in the text since there are mistakes. Check also the format of the list of references.

Responses:

We checked spacing but we will check more when all the editing is accepted and mistakes in spacing become clearer.

We checked all the citations and made corrections; now they all match the reference numbers.

The reference format has been checked and corrected. However, once accepted, we will work with the editing staff for minor changes if required.

Round 3

Reviewer 3 Report (New Reviewer)

I have no major comments

Only few mistakes

Lns 20, 22. Full stops are missing in sentences of the respective lines.

Ln 110. Delete on of the two “the”

Ln 179 insert a “,” in “... Peru, CIP....”

Ln 180. Delete the first “,”

Ln 192. Delete the reference “25”

Ln 210. Replace at the beginning of the paragraph “The study” with “Inglis et al.”

Ln 255. Replace “species” with “strains”

Ln 257. Close the square brackt

Only few mistakes

Lns 20, 22. Full stops are missing in sentences of the respective lines.

Ln 110. Delete on of the two “the”

Ln 179 insert a “,” in “... Peru, CIP....”

Ln 180. Delete the first “,”

Ln 192. Delete the reference “25”

Ln 210. Replace at the beginning of the paragraph “The study” with “Inglis et al.”

Ln 255. Replace “species” with “strains”

Ln 257. Close the square brackt

Author Response

Response to the Reviewer’s suggestions on November 16, 2024:

Dear Reviewer,

We thank you once again for indicating the typos and other suggestions to improve the manuscript. We have incorporated those suggestions and highlighted the respective areas. Regarding the clarity of Figure 1, we have incorporated the pdf files of the respective Figures. The fonts of Fig 1 could be enlarged if the printing department would split Fig 1 into two pages.

Regards.

Lns 20, 22. Full stops are placed at the ends of respective lines.

Ln 110. Deleted one of the two “the”

Ln 179 Inserted a “,” in “... Peru, CIP....”

Ln 180. Deleted the first “,”

Ln 192. Deleted the reference “25”

Ln 210. Replaced at the beginning of the paragraph “The study” with “Inglis et al.”

Ln 255. Replaced “species” with “strains”

Ln 257. Closed the square bracket.

This manuscript is a resubmission of an earlier submission. The following is a list of the peer review reports and author responses from that submission.

Round 1

Reviewer 1 Report

This study phylogenetically re-analyzed 44 (??) strains identified as Trichoderma harzianum in three (or five?) studies from the soil of South and Central America, based on tef1α, rpb2 and ITS sequences. These strains were identified as 8 (??) species after analyzing. Each of these numbers is fraught with confusion. Please specify extractly in the Abstract and Materials and Methods parts. Moreover, the whole article is very lengthy, please reorganize the article structure and simplify the language.

Also, please explain questions below:

1. How many studies did you selected? Why? Are these the only studies relevant to HCC from the soil of South and Central America? 

2. How many strains were analyzed in this study, and how many species were identified?

3. Why are you doing this research? Just want to know how many HCC species in the soil of South and Central America?

Line 21-22: Eight species were shown in Figure 3, but only seven species listed here?

Line 11-26: Please add more information about how many strains were analyzed in this study, and how many species were identified.

Line 85-89: Why analyzed these strains here?

Table 1: "Tef1α" should be "tef1α". "*" should be marked after strain number, not species name.

Line 130-136: "It also indicated that 78% of the sequences that were deposited had previously been erroneously identified". However, in this study, you analyzed the first 100 hit sequences. The conclusion should be wrong.

Line 147-161: How many HCC species did you ananlyzed?

Figure 2: in the figure caption, you indicated this tree was performed by tef1α-rpb2-ITS. However, in the tree, only one GenBank accession number were shown. Why?

Reviewer 2 Report

In this article  the Authors re-classified the isolates of the Trichoderma harzianum complex obtained from five previous surveys in Central and South America using both  BLAST search in NCBI database and phlogenetic analysis of three loci ITS, tef1alfa and rpb2. the study evidentiated several previous misidentifications and was useful to properly identify the prevalent Trichoderma species in this continent and their proportions. This study paves the way for similar investigations in other geographical areas  in the world . Moreover the precise and correct identification of Trichoderma species  according to the criteria of modern molecular taxonomy is essential for the the application of Trichoderma as BCA, plant growth promoter and bioremediation agent as well as to develop commercial products based on this fungus as active ingredient.

I only added minor comments, suggestions and corrections, as follows:

- Line 37. Extend the concept of Trichoderma as a multipurpose microrganism including the application of this fungus in  environmental remediation (two references are suggested).

- Line 282. References are best cited at the end of a sentence.

- Line 285. The correct name of the species is T. lentiforme.

- Line 310. I warmly suggest to modify the sentence as indicated (seed attached PDF file) 

For other minor text editings and suggestions see the notes in the text  (attached PDF file)

Reviewer 3 Report

The article is a simple study mainly about the re-identification of Trichoderma strains from the Harzianum Complex clade (HCC). The title refers to the distribution of HCC species in Central and South America, however the robustness of the work lies in the re-identification. The authors should include this in the title of the article.

Second, the introduction is long and contained banal information. I recommend including more bibliography about the current distribution of Trichoderma strains in Central and South America and delete the generalities about the topic. There is unnecessary and repeated information. In addition, I think that the hypothesis and objective of the research work are not clearly written. The introduction should be rewritten.

Third, the results section is difficult to read; it includes some discussions that should be written in the Discussion section.

In the discussion, the authors discuss the presence of Trichoderma species in the world and I think that this does not reflect the results obtained in the research work.

I think again that the hypothesis and the objectives are not clear in the research work, which makes it difficult to write a discussion and makes it difficult to read the article.

There are spelling errors in the names of the Trichoderma species or words as "stain" (is strain?) in all the article. Some exemples: 

-Line 244: T. Hortense

-Line 285: "Lentifore"?

The format of the tables is difficult to read. I recommend re-written.